# A Study on Dual-Gate Dielectric Face Tunnel Field-Effect Transistor for Ternary Inverter

**DOI:** 10.3390/nano14151307

**Published:** 2024-08-03

**Authors:** Aoxuan Wang, Hongliang Lu, Yuming Zhang, Jiale Sun, Zhijun Lv

**Affiliations:** Key Laboratory for Wide Band Gap Semiconductor Materials and Devices of Education Ministry, School of Microelectronics, Xidian University, Xi’an 710071, China; 24111110087@stu.xidian.edu.cn (A.W.); zhangym@xidian.edu.cn (Y.Z.); jialesun@stu.xidian.edu.cn (J.S.); zhijjunlv@stu.xidian.edu.cn (Z.L.)

**Keywords:** ternary inverter, multiple subthreshold voltage (multi-VTH), tunnel field-effect transistor (TFET), static noise margins (SNMs)

## Abstract

In this article, we propose a dual-gate dielectric face tunnel field-effect transistor (DGDFTFET) that can exhibit three different output voltage states. Meanwhile, according to the requirements of the ternary operation in the ternary inverter, four related indicators representing the performance of the DGDFTFET are proposed, and we explain the impact of these indicators on the inverter and confirm that better indicators can be obtained by choosing appropriate design parameters for the device. Then, the ternary inverter implemented with this device can exhibit voltage transfer characteristics (VTCs) with three stable output voltage levels and bigger static noise margins (SNMs). In addition, by comparing the indicators of the DGDFTFET and a face tunnel field-effect transistor (FTFET), as well as the SNM of inverters, it is demonstrated that the performance of the DGDFTFET far surpasses the FTFET.

## 1. Introduction

Over the past several decades, owing to the continual miniaturization of CMOS devices, the leakage current has increased rapidly because of the short-channel effects (SCEs) and the increase in the interconnect density. To reduce the power density in very-large-scale integration (VLSI), a vast number of studies have focused on multi-valued logic, especially standard ternary inverters (STIs). As a critical building block of the ternary system, the ternary inverter has three stable states compared to the conventional inverter, thereby enabling higher information density and fewer interconnects [1]. However, some standard ternary inverters consist of multiple conventional CMOS devices [2,3,4]. These approaches lead to an increase in the number of devices and power density for the elementary ternary inverter, which is contrary to the original purpose of multi-valued logic (MVL), which is to reduce the number of transistors and power density.

Recently, various multi-valued logic (MVL) unit devices have been proposed, and tunnel FETs (TFETs) are one of them. TFETs have been extensively studied since they can break through the limitations of 60 mV/decade at room temperature, which is based on a band-to-band tunneling (BTBT) mechanism and is widely used in low-power applications [5,6,7,8,9,10,11,12]. Some studies have been implemented, showing that STIs based on TFETs can be achieved [13,14,15,16,17]. The transfer characteristics of these devices include the constant gate voltage, independent current region, and gate voltage-dependent exponential current region; the ternary operation can be achieved by matching the constant current region. However, owing to the large off-state current, the static power consumption of the STIs consisting of these devices is very high. Zhijun et al. proposed the FTFET, which has multi-V_TH_ devices [18] (as shown in Figure 1a), where the static power consumption is significantly reduced due to the smaller off-state current. However, the unstable characteristics in the FTFET result in the STI, which is that this device has an unstable middle state and a small SNM. Therefore, it is crucial that we improve the FTFET to obtain a high performance and further improve the SNM of the STI. The indicators of multi-V_TH_ devices should be proposed and discussed. By comparing these indicators of the improved transistor with the FTFET, we confirm the advantages of the new device.

In this paper, the dual-gate dielectric face tunnel field-effect transistor (DGDFTFET) is demonstrated. Meanwhile, the related indicators, which represent the performance of the DGDFTFET, are proposed according to the requirements of the ternary operation, and we explained the impact of these indicators on the device and ternary inverter. By comparing the performance indicators of the DGDFTFET with the FTFET at the device level and circuit level, we demonstrated the advantages of the dual-gate dielectric structure. In addition, it is confirmed that the high-performance device indicators and ideal ternary operation in the STI can be obtained by choosing an appropriate source doping and bias voltage.

## 2. Device Structure

The schematic device structure proposed in this article is shown in Figure 1b. The design parameters of the MFTFET are similar to the design parameters of the DGDFTFET, taking the n-type DGDFTFET as an example. The length of the source region (L_source_), channel region (L_channel_), and drain region (L_drain_) are 150 nm, 100 nm, and 50 nm, respectively. The channel thickness is 40 nm. In terms of structure, the FTFET has a gate-to-source overlap region with a distance of L_overlap_ (L_overlap_ = 50 nm) compared to the traditional TFET structure, which leads to an abrupt increase in drain current, thus forming the second subthreshold voltage. The DGDFTFET proposed in this paper has the structure of a dual-gate dielectric model compared with the FTFET, which has a single-material gate (see Figure 1). The material of the gate oxide between the gate and source is SiO_2,_ and the gate oxide between the gate and channel uses HfO_2_ as its material to open the tunnel faster. The concentration of the heavily P-doped source region is 5 × 10^19^ cm^−3^, the doping type of the drain region is N-type with a doping concentration of 2 × 10^19^ cm^−3^, and the uniform doping profile of the N-type channel region is 1 × 10^16^ cm^−3^. The important design parameters are shown in Table 1.

Numerical simulations have been performed by using the Technology Computer-Aided Design (TCAD) tool (Synopsys Sentaurus). The dynamic nonlocal BTBT model which can properly state the tunneling mechanism of the TFET is adopted. The bandgap narrowing model, Shockley–Read–Hall (SRH) recombination, Fermi statistics, and doping-dependent mobility model are also used to simulate this model. Figure 2 indicates the comparison of simulated transfer characteristics curves and the measurement results of the device in Ref. [19]; they have a good match, which demonstrates the suitability of the simulation model.

## 3. Basic Properties of the DGDFTFET

The indicators which can represent the performance of the DGDFTFET are discussed. It is obvious from Figure 3 that the device has two average subthreshold swings; the first average subthreshold swing is defined as SS_1_ which can measure the rate of the drain current transitions from the off current to the first low transconductance region, and the drain current transition from the first low transconductance region to the second low transconductance region is affected by the second average subthreshold swing SS_2_; the transition of the current are steeper with the decrease in SS in the device, which means, when the transition of a different state in the ternary inverter is steeper, a bigger SNM is obtained easily. In addition, a smaller subthreshold swing makes it easier to achieve V_DD_ scaling and further reduce power consumption [20,21]. Therefore, the SS of the device which has a better performance should be smaller; the SS_1_ and SS_2_ can be calculated as:(1)$${SS}_{1}=\frac{dV_{G}}{d{Iog}^{I_{D}}}=\frac{V_{lt1}-V_{TH1}}{{Iog}^{I_{lt1}}-{Iog}^{I_{TH1}}}$$
(2)$${SS}_{2}=\frac{dV_{G}}{d{Iog}^{I_{D}}}=\frac{V_{lt2}-V_{TH2}}{{Iog}^{I_{lt2}}-{Iog}^{I_{TH2}}}$$
where V_TH1_ is the line-tunnel subthreshold voltage, and V_TH2_ is the face-tunnel subthreshold voltage, with the corresponding drain current I_TH1_ and I_TH2_, respectively. The V_lt1_ is the corresponding gate voltage when the line-tunnel current enters the first low transconductance region; the corresponding drain current is Ilt1; the voltage V_lt2_ is the corresponding gate voltage when the face-tunnel current enters the second low transconductance region; the corresponding drain current is I_lt2_; and the slope of the four points in the curve is 1 (as shown in Figure 3). In addition, the flatness of the line-tunnel current in the low transconductance region is also an important indicator which can reflect the impact of the gate voltage on the channel, by using the transconductance ($g_{m\_min1}$) to represent this indicator. A smaller $g_{m\_min1}$ means that the drain current is almost independent of the gate voltage; the I_D_ hardly changes with the increase of the V_G_, which makes the form of the intermediate state in the ternary inverter easier, and obtains a larger SNM. Therefore, the device should have a small $g_{m\_min1}$; the $g_{m\_min1}$ can be calculated as:(3)$$g_{m\_min1}=\frac{dI_{D}}{dV_{G}}$$

Meanwhile, as an important parameter of the inverter, the width of the intermediate state in the STI can be influenced by the width ($∆V_{L}$) of the first low transconductance region (as shown in Figure 3); a stable intermediate state and an appropriate SNM can be obtained by using a suitable intermediate state width. Therefore, the controllability of $∆V_{L}$ is important. The $∆V_{L}$ can be calculated as:(4)$$∆V_{L}=V_{TH2}-V_{lt1}$$

Next, the formation mechanism of three tunneling currents is discussed. When the gate voltage (V_G_) of the N-type DGDFTFET is less than V_TH1_, there is no charge tunneling in the TFET; therefore, the tunneling current is not generated, and the device is in the off state at this time. As the gate voltage increases to a certain value (V_TH1_ < V_G_ < V_TH2_), the conduction band in the channel is located below the valence band of the source region (as shown in Figure 4a), charges can tunnel from the source to the channel and are finally collected by the drain, the line-tunnel current is formed, and the device enters the line-tunnel region. Note that a flat current (low transconductance region) occurs in the line-tunnel region; the reason for this phenomenon is related to the electron screening. The energy barrier between the channel and drain decreases with the increase in gate voltage, and, finally, charges of the drain region are injected to the channel region, which results in the electrostatic potential of the channel region being almost unaffected by the gate voltage. Therefore, it results in a stable line-tunnel current. Until the gate voltage exceeds V_TH2_, the conduction band edge of the source surface under the gate-to-source overlap is located below the valance band (see Figure 4b), the face-tunnel energy window along the *y*-axis appears under the gate-to-source overlap region, and the face-tunnel current is generated. At present, the total current consists of the line-tunnel current and face-tunnel current, which lead to an abrupt increase in the drain current, and the device enters the face-tunnel region.

The advantages of the dual-gate dielectric structure are introduced. By calculating from Figure 3, the SS_1_ of the DGDFTFET is 37.6 mV/dec, the SS_1_ of the FTFET is 59.4 mV/dec, and the line-tunnel opens earlier in the DGDFTFET. This is due to the HfO_2_, which has an electrostatic characteristic of high-k, being used as the material of the gate oxide between the gate and channel (see Figure 1b); it is obvious from Figure 5a that the line tunneling only occurs in the DGDFTFET at V_G_ = 0.25 V under the value of V_D_ being 0.5 V, which makes the line-tunnel open earlier, resulting in the DGDFTFET having a smaller V_TH1_. Figure 5b indicates the comparison of the two devices in surface potential and BTBT generation; it is obvious that the DGDFTFET has a stronger gate control ability and a larger BTBT generation under the same V_G_; therefore, a smaller SS_1_ is obtained in the DGDFTFET. Similarly, the SS_2_ of the DGDFTFET and the FTFET is 60.2 mV/dec and 60.6 mV/dec, respectively. Owing to the unchanged material between the gate and source, the SS_2_ of the two devices are almost equal. Meanwhile, the $g_{m\_min1}$ of the DGDFTFET can be calculated as 0.8 nS, the $g_{m\_min1}$ of the FTFET is 5.36 nS, and the $∆V_{L}$ of the DGDFTFET and the FTFET are 0.86 V and 0.12 V, respectively. This is due to the DGDFTFET having a smaller V_TH1_ and SS_1_, which makes the line tunneling of the DGDFTFET occur earlier and enter the low transconductance region faster; therefore, a smaller $g_{m\_min1}$ and a larger $∆V_{L}$ are obtained in the DGDFTFET. To sum up, the performance of the FTFET which has a dual-gate dielectric structure is better than single-material-gate dielectric FTFET.

Then, two main parameters which can influence these indicators in the device are discussed. Figure 6a,b indicates the change in the transfer characteristics and two performance indicators under different source doping (N_source_). The line-tunnel current and $∆V_{L}$ become larger with the N_source_ increasing as shown in Figure 6a; this is due to the larger doping of the source region resulting in more charges being able to tunnel from the source to the channel, which leads to the larger drain current. Meanwhile, because of the higher N_source_, the face-tunnel window becomes more difficult to open; consequently, a larger V_TH2_ and $∆V_{L}$ are obtained. Figure 6b indicates a comparison of performance parameters between two devices; the $∆V_{L}$ in the FTFET hardly changes with the increase in N_source_ compared to the case of the $∆V_{L}$ in the DGDFTFET increasing with N_source_. In addition, the $g_{m\_min1}$ of both TFETs becoming larger with the increase in N_source_ owing to the increasing of I_D_ results in the larger I_D_ difference, and the $g_{m\_min1}$ of the DGDFTFET is much lower than the FTFET under a different N_source_. Figure 6c indicates the transfer characteristic curves of the DGDFTFET at various drain voltages (V_D_); the low transconductance region becomes flatter and wider with the decrease in V_D_. This is due to the energy barrier between the channel and drain reducing as V_D_ decreases, which results in more screening electrons; thus, the line-tunnel current occurs in a smaller gate voltage and the width of the low transconductance region becomes wider with the V_D_ decreasing. Figure 6d shows the comparison of indicators between two devices: compared to the case of $∆V_{L}$ in the FTFET almost not changing with V_D_, $∆V_{L}$ of the DGDFTFET decreases with V_D_. In addition, owing to the line-tunnel current becoming steeper with increases in V_D_, the $g_{m\_min1}$ of them increases with V_D_; however, the variation amplitude and the value of the DGDFTFET are much lower. To sum up, it is obvious that the $∆V_{L}$ of the DGDFTFET has a stronger controllability. Meanwhile, the technological parameters and bias voltage, such as the N_source_ or V_D_, must be chosen carefully to obtain the appropriate device indicators, which further helps us obtain a higher inverter performance.

## 4. Ternary Inverter Optimization and Comparison

Figure 7 shows the voltage transfer characteristics (VTCs) of the standard ternary inverter (STI) obtained using the DGDFTFET as mentioned in Figure 3. The voltage V_DD_ of the ternary inverter is 2 V. We define several important voltages related to the STI to make it easier to explain how to form the ternary inverter; these parameters correspond to the points in the VTC curve with a slope of −1. The voltage VIL is the maximum input voltage (V_in_) when the corresponding input logic of the STI is “0”; the corresponding output voltage (V_out_) is V_OH_ which can be treated as logic “2”. Similarly, the voltage V_IH_ can be regarded as the minimum input voltage when the input logic is “2”; the corresponding output voltage is V_OL_ which can be treated as logic “0”. The voltage V_IML_ and V_IMH_ are the two intermediate input voltages when the inverter output logic is “1”, with the corresponding output voltages (logic “1”) V_OMH_ and V_OML_.

Based on the transfer characteristic curves at various V_D_ (see Figure 6c), the VTC of the inverter can be easily expected, and the pull-down and pull-up device can be simplified as variable resistors to explain the formation mechanism of the ternary operation; it can be explained as follows.

(1)As the input voltage (V_in_) is close to 0 V, the pull-up device is in the face-tunnel current region (small equivalent resistance), and the pull-down device is in the off state (large equivalent resistance). At this point, the V_out_ is close to V_DD_, which corresponds to the output logic of “2”.(2)When the input voltage becomes closer to half of the V_DD_ (V_IML_ < V_in_ < V_IMH_), both the pull-up device and the pull-down device reach the line-tunnel region and start to have a similar equivalent resistance. Considering the voltage-division effect, the intermediate state is formed (V_out_ ≈ V_DD_/2).(3)When the input voltage exceeds V_IH_, the n-type device reaches the face-tunnel region (small equivalent resistance), whereas the p-type DGDFTFET enters the off state (large equivalent resistance), and V_out_ starts to converge to 0 V. At this point, a voltage corresponding to the output logic of “0” is obtained.

SNM is an important indicator for measuring the performance of the STI. Figure 8 shows the butterfly curve of the ternary inverter; in contrast with the binary inverter with only two noise margins, the ternary inverter consists of four noise margins, and the SNM is obtained by extracting the shortest diagonal of the four squares inscribed inside the butterfly curve. Meanwhile, the width of the intermediate state in the STI has a significant impact on the SNM; some research suggests that the width of the intermediate state in the STI should be as close as possible to V_DD_/2 to ensure the maximum SNM [22,23]. In addition, the symmetry of the VTC curve is also an important factor affecting the SNM. To ensure that the width of the intermediate state is V_DD_/2 and to ensure the symmetry of the VTC curve, two points need to be noted: they are V_M1_ and V_M2_, respectively. In the binary inverter, V_M_ is the point on the VTC curve where V_out_ = V_in_; the V_M_ should be as close as possible to (1/2V_DD_, 1/2V_DD_) to make the two noise margins equal, and obtain the larger NM. But, in the ternary inverter, the voltage V_M1_ and V_M2_ must be on the point (1/4V_DD_, 1/2V_DD_) and (3/4V_DD_, 1/2V_DD_) to ensure the maximum SNM. The V_M1_ is the point on the VTC curve where V_out_ = 2V_in_; and V_M2_ is the point on the VTC curve where V_out_ = 2/3V_in_.

Figure 9a shows the voltage transfer characteristics of the STI based on the DGDFTFET and FTFET under optimal design parameters. It is obvious that the V_out_ transition (from logic ‘2’ to logic ‘1’ and from logic ‘1’ to logic ‘0’) in the STI consists of the fact that the DGDFTFET is steeper than the FTFET-based STI. The reason for this phenomenon can be explained by equivalent resistance: a smaller SS_1_ means the difference of resistance can quickly transition from being larger to almost equal in the transition region; therefore, the STI consists of the DGDFTFET having a steeper Vout transition. In addition, the V_M1_ of the two inverters are (0.54, 1.1) and (0.7, 1.3), respectively. The V_M2_ of the two inverters are (1.46, 0.9) and (1.27, 0.74), respectively. It is obvious that the V_M1_ and V_M2_ of the STI based on the DGDFTFET are closer to the ideal value, and the intermediate state of this inverter is more stable. The parameters $g_{m\_min1}$ and $∆V_{L}$ are the main affecting factors: a smaller $g_{m\_min1}$ is indicated in the DGDFTFET, which makes the n/p-type device have a good matching in the line-tunnel region; therefore, the equivalent resistance of the two devices in the line-tunnel region is almost equal. Therefore, the intermediate state in the STI consists of the DGDFTFET being more stable, and the V_M1_ and V_M2_ in the STI consist of the DGDFTFET being closer to the ideal value. Meanwhile, the parameters $∆V_{L}$ can also influence the V_M1_ and V_M2_, $∆V_{L}$ can influence the width of the intermediate state in the STI, and a suitable $∆V_{L}$ make the V_M1_ and V_M2_ closer to the ideal value on the *x*-axis. The butterfly curves of the two devices are shown in Figure 9b, the SNM of the two inverters is obtained by extracting from the VTC of the STI, the SNM of the inverter consists of the DGDFTFET being able to reach 421.1 mV at a supply voltage of 2 V, the SNM of the FTFET-based STI under the same design parameters is only 23.2 mV, and the SNMs of the two inverters have a huge gap.

Next, the impact of N_source_ and V_DD_ on the inverter is discussed. Figure 10a indicates that the VTC of the STI corresponds to different N_source_; the width of the intermediate state ($∆V_{IM}$) becomes larger with the increase in N_source_. This is due to the $∆V_{L}$ increasing with N_source_; a larger matching region is obtained. By extracting the SNM from the VTC of the STI, Figure 10b indicates that the maximum value of the SNM in the DGDFTFET appears when the N_source_ reaches 8 × 10^19^ cm^−3^ (SNM = 421.1 mV); regardless of the N_source_ being higher or lower than 8 × 10^19^ cm^−3^, the SNM significantly decreases. Figure 10c indicates the VTC of the STI corresponding to different V_DD_; it can be seen that a stable ternary operation can be formed over a wide supply voltage range. In addition, the SNM is obtained for different V_DD_; it can be seen from Figure 10d that the optimal SNM of the DGDFTFET occurs at V_DD_ = 2.0 V, according to the previous analysis; in this case, the width of the intermediate state in the STI is closest to V_DD_/2 and the voltage V_M1_ and V_M2_ are closest to the ideal value which can make the inverter obtain the biggest SNM. Hence, the N_source_ and V_DD_ can effectively regulate the ternary operation and the SNM of the STI, thus providing a technological approach for the optimization design of the ternary inverter.

As mentioned above, the relevant device and inverter performance indicators are displayed in Table 2. By comparing the performance indicators in the device and the SNM of the STI, it is obvious that the inverter consists of the DGDFTFET having better performance indicators. Therefore, the FTFET which has a dual-gate dielectric structure is more suitable to be applied to the ternary inverter compared to the single-material-gate FTFET. Meanwhile, the appropriate optimization can make the STI achieve the ideal ternary operation and obtain a larger SNM.

## 5. Conclusions

In this study, on the basis of the face tunnel field-effect transistor (FTFET), the FTFET which has a dual-gate dielectric structure is introduced; three different regions of tunneling currents are displayed in the DGDFTFET which can form the ternary inverter with a stable intermediate state. Meanwhile, the related performance indicators of the DGDFTFET are proposed; it is revealed that SS_1_, SS_2_, $∆V_{L}$ and $g_{m\_min1}$ are important indicators for the device to affect the stability of the ternary operation and SNM. By comparing the two devices at the device level and circuit level, it is demonstrated that the FTFET of the dual-gate dielectric structure has better performance indicators, and it is more suitable to be applied to the ternary inverter compared to the single-material-gate FTFET. In addition, it is confirmed that the performance indicators can be affected by the device parameters or bias voltage, such as source doping and V_DD_, which further influence the performance of the STI. An appropriate optimization of the source doping or bias voltage can obtain the stability of the ternary operation and a large SNM.

## Figures and Tables

**Figure 1 nanomaterials-14-01307-f001:**
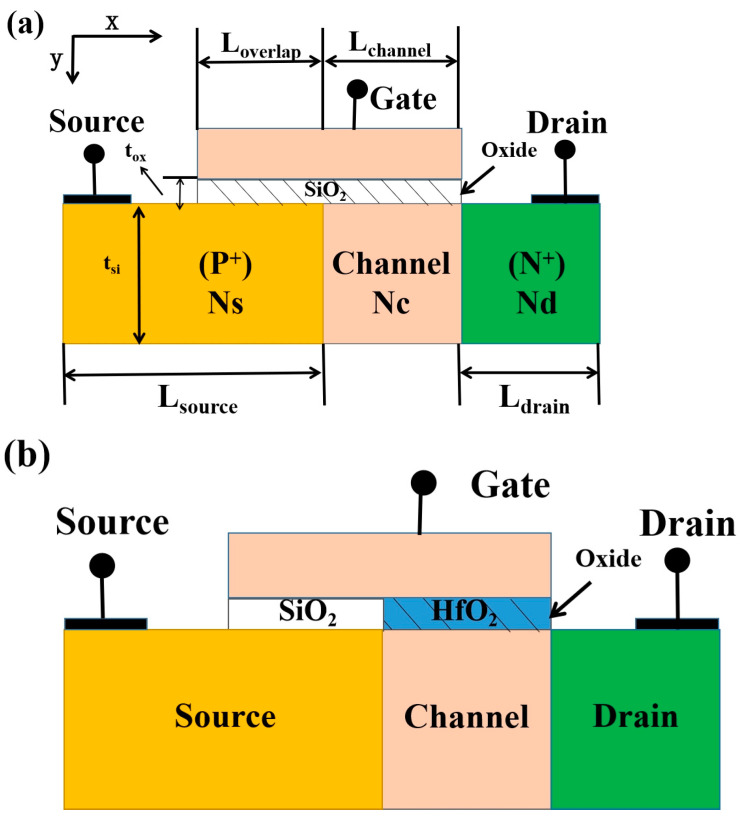
Schematic cross-sectional view of the (**a**) FTFET proposed in [16] and (**b**) DGDFTFET.

**Figure 2 nanomaterials-14-01307-f002:**
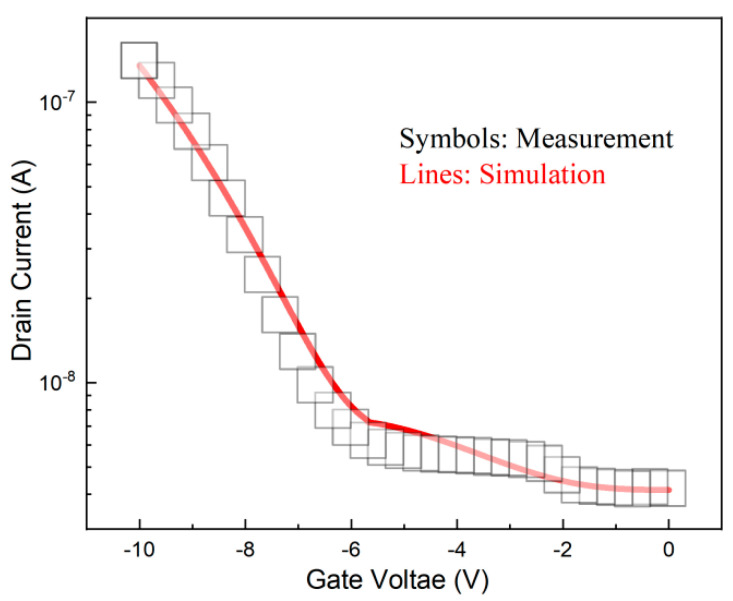
Calibration of simulation model for Si TFET. The measurement data were taken from [18], Figure 5a.

**Figure 3 nanomaterials-14-01307-f003:**
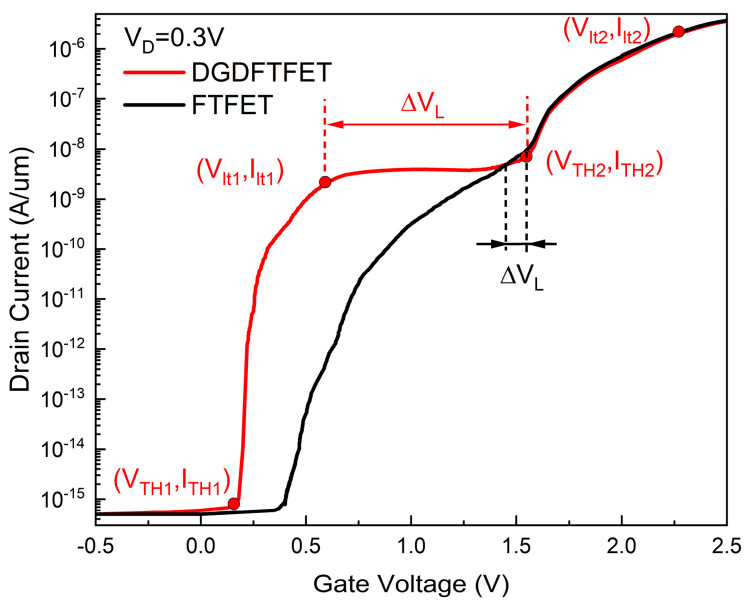
Transfer characteristics of the n-type DGDFTFET and FTFET at V_D_ = 0.5 V.

**Figure 4 nanomaterials-14-01307-f004:**
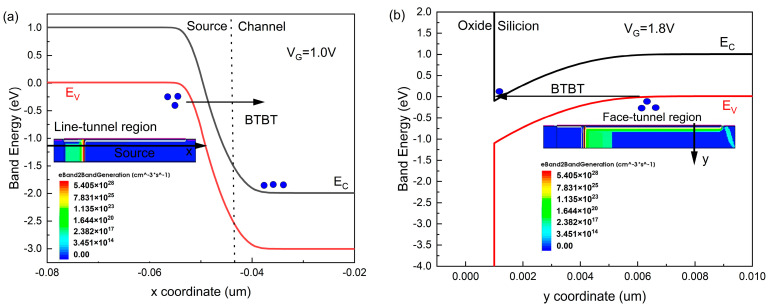
Band diagrams along x at (**a**) line-tunnel region (V_G_ = 1.0 V); and (**b**) face-tunnel region (V_G_ = 1.8 V).

**Figure 5 nanomaterials-14-01307-f005:**
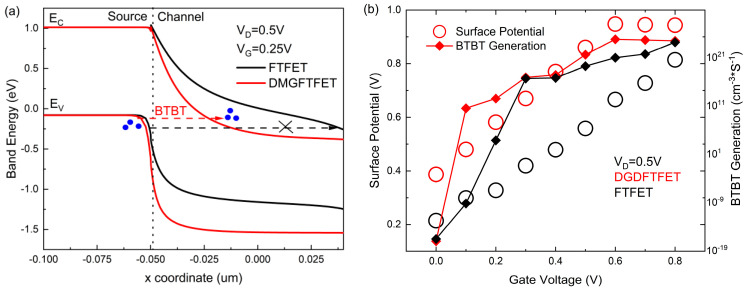
(**a**) Band diagrams of the two devices along x at line-tunnel region (V_G_ = 0.25 V); and (**b**) for V_G_ changes of surface potential and BTBT generation in DGDFTFET and FTFET.

**Figure 6 nanomaterials-14-01307-f006:**
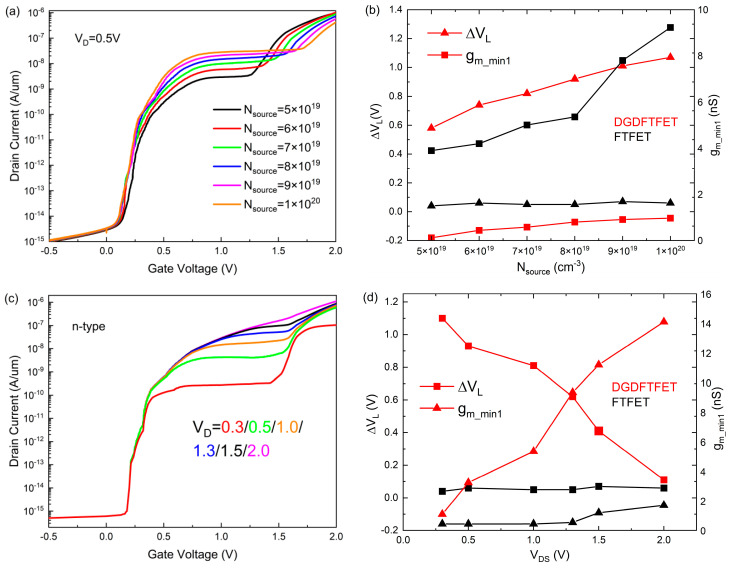
The change of (**a**) transfer characteristic curves of DGDFTFET with respect to N_source_; (**b**) $∆V_{L}$ and and $g_{m\_min1}$ under different N_source_; (**c**) transfer characteristic curves of DGDFTFET with respect to V_D_; and (**d**) $∆V_{L}$ and $g_{m\_min1}$ under different V_D_.

**Figure 7 nanomaterials-14-01307-f007:**
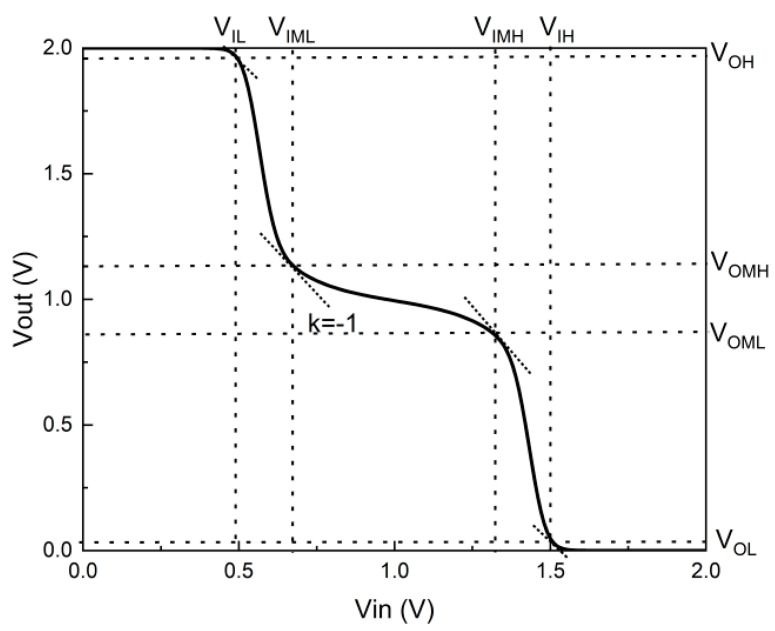
DGDFTFET exhibiting ternary inverter VTC.

**Figure 8 nanomaterials-14-01307-f008:**
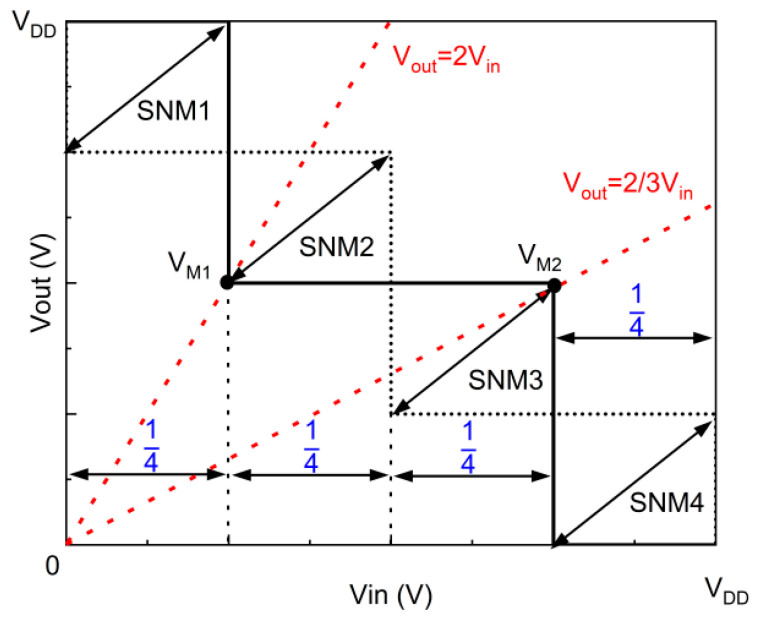
Butterfly curve for STI for which the intermediate region has a width of V_DD_/2.

**Figure 9 nanomaterials-14-01307-f009:**
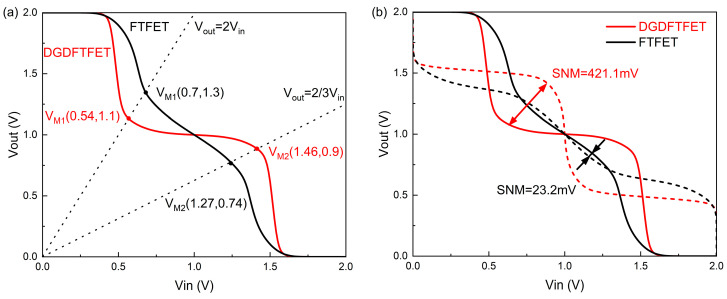
(**a**)The VTC of the inverter consists of DGDFTFET and FTFET under the same design parameters; and (**b**) butterfly curve for DGDFTFET and FTFET ternary VTC.

**Figure 10 nanomaterials-14-01307-f010:**
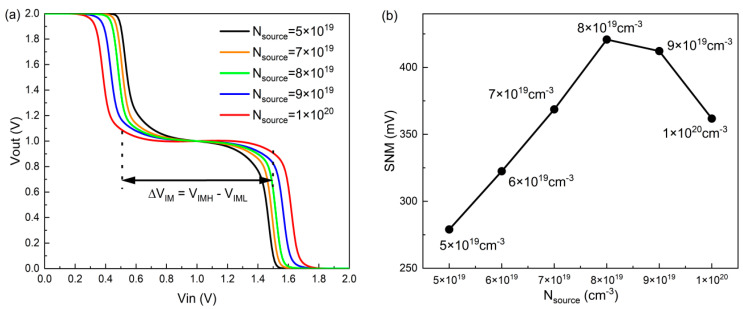
The change in (**a**) VTC curves of STI based on DGDFTFET with respect to N_source_; (**b**) SNM under different N_source_; (**c**) VTC curves of STI based on DGDFTFET with respect to V_D_; and (**d**) SNM under different V_D_.

**Table 1 nanomaterials-14-01307-t001:** Device parameter.

Definition	Symbol	Value
Supply voltage (V)	V_DD_	2
Source length (nm)	L_source_	150
Channel length (nm)	L_channel_	100
Drain length (nm)	L_drain_	50
Equivalent oxide thickness (nm)	EOT	1.0
Gate-to-source overlap length (nm)	L_overlap_	50
Source doping (atoms/cm^3^)	N_S_	5 × 10^19^
Channel doping (atoms/cm^3^)	N_C_	1 × 10^16^
Drain doping (atoms/cm^3^)	N_d_	2 × 10^19^
Gate workfunction (eV)	WF	4.3

**Table 2 nanomaterials-14-01307-t002:** Indicators of performance.

Indicators of Device Performance	SS_1_ (mv/dec)	SS_2_ (mv/dec)	$∆V_{L}$ (V)	$g_{m\_min1}$ (nS)
FTFET	37.6	60.6	0.12	5.36
DGDFTFET	59.4	60.2	0.86	0.8
Indicators of STI performance	V_M1_		V_M2_	SNM (mV)
Based on FTFET	(0.7, 1.3) (0.54, 1.1)		(1.27, 0.74) (1.46, 0.9)	23.2 421.1
Based on DGDFTFET				

## Data Availability

The data presented in this study are available upon request from the corresponding author. The data are not publicly available due to confidentiality of the project.

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
