# Peer review of "A Study on Dual-Gate Dielectric Face Tunnel Field-Effect Transistor for Ternary Inverter"

_nanomaterials, 2024, doi:10.3390/nano14151307_

Round 1
Reviewer 1 Report
Comments and Suggestions for Authors
The merit of TFET is the fast turn on by sharp subthreshold slope. This study shows low voltage operation in the first turn on region and becomes the same as conventional TFET. The other new finding in this device is the ternary inverter.
Here are the comments:
I guess the low voltage operation is due to the extra high-k HfO2. Will the threshold voltage and on-current be improved by using the single high-k HfO2? Please notice that the crucial factor for TFET is the high on-current under low voltage operation.
The title may be better to change the “Dual Material Gate” to “Dual Gate Dielectric”.
Please also notice that the gate length is the smallest dimension in the lithography process. It will be hard to use the SiO2 and HfO2 gate dielectric under the metal gate. Please give a possible explanation.
Comments on the Quality of English Language
It will be better if English writings could be improved.
Reviewer 2 Report
Comments and Suggestions for Authors
In this study, the author propose a dual material gate face tunnel field-effect transistor (DMGFTFET) that can achieve three distinct output voltage states. We introduce four key performance indicators relevant to ternary operations in ternary inverters, detailing their impact on inverter performance. By optimizing device design parameters, the DMGFTFET-based ternary inverter exhibits voltage transfer characteristics with three stable output levels and improved static noise margins (SNM). Comparative analysis shows that DMGFTFET outperforms the traditional face tunnel field-effect transistor (FTFET) in terms of these indicators and SNM, demonstrating superior overall performance.
The manuscript is designed well, but some issues should be addressed to improve its quality as suggested below,
I. Introduction section does not provide a clear picture of the gap and need of this study and the issues that need to be addressed. The authors should provide some more details and should remove unnecessary information. So please clearly state the objective of this study shortly in the introduction section.
II. The more important concern is about the estimation of the SS values extracted in this manuscript. The author should add more details about the estimation of SS values and should also compared these results with the recent reports.
III. In Figure 4b anc Fig 4c, authors mentioned about the BTBT, but I think it would be reasonable to consider it at different temperatures for more accurate description.
IV. The introduction should contain some potential applications of the 2D materials based FET device as given in these reports, (https://doi.org/10.1016/j.diamond.2024.111089), (https://doi.org/10.1016/j.nanoen.2023.109106), which can be added in the introduction section.
V. Minor language changes need to be addressed before submitting the revised version of the manuscript
Remarks: Major Revision is Required to improve the manuscript quality
Comments on the Quality of English LanguageEnglish language should be improved.
Round 2
Reviewer 1 Report
Comments and Suggestions for Authors
The authors have answered my previous comments.
Reviewer 2 Report
Comments and Suggestions for Authors
Accept in present form